# Dielectric, Mechanical, and Thermal Properties of Crosslinked Polyethylene Nanocomposite with Hybrid Nanofillers

**DOI:** 10.3390/polym15071702

**Published:** 2023-03-29

**Authors:** Nurul Iman Abdul Razak, Noor Izyan Syazana Mohd Yusoff, Mohd Hafizi Ahmad, Muzafar Zulkifli, Mat Uzir Wahit

**Affiliations:** 1Faculty of Chemical and Energy Engineering, Universiti Teknologi Malaysia, Johor Bahru 81310, Johor, Malaysia; 2Advanced Membrane Technology Research Centre (AMTEC), Universiti Teknologi Malaysia, Johor Bahru 81310, Johor, Malaysia; 3Institute of High Voltage and High Current, School of Electrical Engineering, Universiti Teknologi Malaysia, Johor Bahru 81310, Johor, Malaysia; 4Green Chemistry and Sustainability Cluster, Branch Campus, Malaysian Institute of Chemical and Bioengineering Technology, Universiti Kuala Lumpur, Taboh Naning, Alor Gajah 78000, Melaka, Malaysia; 5Centre for Advanced Composite Materials (CACM), Universiti Teknologi Malaysia, Johor Bahru 81310, Johor, Malaysia

**Keywords:** crosslinked polyethylene, hybrid filler, dielectric properties, mechanical properties, thermal properties, hybrid nanocomposite

## Abstract

Crosslinked polyethylene (XLPE) nanocomposite has superior insulation performance due to its excellent dielectric, mechanical, and thermal properties. The incorporation of nano-sized fillers drastically improved these properties in XLPE matrix due to the reinforcing effect of interfacial region between the XLPE–nanofillers. Good interfacial strength can be further improved by introducing a hybrid system nanofiller as a result of synergistic interaction between the nanofiller relative to a single filler system. Another factor affecting interfacial strength is the amount of hybrid nanofiller. Therefore, the incorporation amount of hybridising layered double hydroxide (LDH) with aluminium oxide (Al_2_O_3_) nanofiller into the XLPE matrix was investigated. Herein, the influence of hybrid nanofiller content and the 1:1 ratio of LDH to Al_2_O_3_ on the dielectric, mechanical, and thermal properties of the nanocomposite was studied. The structure and morphology of the XLPE/LDH-Al_2_O_3_ nanocomposites revealed that the hybridisation of nanofiller improved the dispersion state. The dielectric, mechanical, and thermal properties, including partial discharge resistance, AC breakdown strength, and tensile properties (tensile strength, Young’s modulus, and elongation at break) were enhanced since it was influenced by the synergetic effect of the LDH-Al_2_O_3_ nanofiller_._ These properties were increased at optimal value of 0.8 wt.% before decreasing with increasing hybrid nanofiller. It was found that the value of PD magnitude improvement went down to 47.8% and AC breakdown strength increased by 15.6% as compared to pure XLPE. The mechanical properties were enhanced by 14.4%, 31.7%, and 23% for tensile strength, Young’s modulus, and elongation at break, respectively. Of note, the hybridisation of nanofillers opens a new perspective in developing insulating material based on XLPE nanocomposite.

## 1. Introduction

Polymeric materials have been well known for years as electrical insulating materials because they have good dielectric, mechanical, and thermal strength. Crosslinked polyethylene (XLPE) has the best insulation properties among polymeric materials. XLPE materials are not limited to low-voltage and medium-voltage cable application, but are also used in high-voltage and extra-high-voltage cables [1,2]. As the XLPE material is subjected to the degradation process caused by high voltage current, it is also exposed to mechanical damage that would occur during installation or operation. Furthermore, the stress and continuous bending of insulating materials would initiate defects and reduce polymeric materials’ durability [3]. Therefore, in addition to dielectric properties, the mechanical and thermal properties of XLPE materials need to be studied to increase its service life by adding nanofiller into XLPE.

Starting with macro-sized fillers, progress in the field has allowed for the expansion to nano-sized fillers. The advantage of nano-sized filler is the high aspect ratio, resulting in high surface area, which can potentially change the property to enhance the insulation system [4]. To utilise the nano-sized filler’s characteristics, uniform distribution of nanofiller into the XLPE matrix needs to be achieved by increasing the interfacial interaction between XLPE–nanofiller and internanofiller. The introduction of a hybrid nanofiller can achieve a robust interfacial bonding compared to a single filler system that is attributed to synergistic effect [5], and this process enhanced the mechanical behaviour of the XLPE nanocomposites [6] in addition to dielectric and thermal properties [7]. The properties of hybrid nanocomposite improved in dielectric properties and thermal conductivity in addition to promoting mechanical properties on Young’s modulus and tensile strength [3,8,9,10] and showed remarkable performance with a homogeneous distribution of hybrid nanofiller obtained at low nanofiller loading with a maximum of 1 wt.% [11,12,13]. These successful studies proved that a combination of two different background characteristics achieve synergistic effect towards the nanocomposite with high compatibility achieved between matrix–nanofiller and interfiller. The nanofillers support each other by bringing the nanotube to platelet [14], inserting the nanosheet into nanofiber [8], and the capability of the nanofiller to sit well with other nanofiller should be considered.

Among the best hybrid nanofiller system selections is layered double hydroxide (LDH). The success of LDH studies on flame retardant, biomedical, gas barriers, and anti-corrosion led to an expansion to electrical insulators [15]. LDH has appreciable thermal stability in improving heat dissipation by distributing the temperature more uniformly throughout the electric insulation cable to ensure the reliability and stability of the nanocomposite material [16,17,18,19]. Moreover, the layered silicate of LDH contributes to increasing most of the mechanical properties, including strength, modulus, and stiffness of nanocomposite. Furthermore, LDH treated with sodium dodecyl sulphate (SDS) would change the morphology, expand the interlayer distance, and allow the polymer to intercalate [20]. With all these advantages, LDH has high potential as an electrical insulator nanofiller to hybrid with aluminium oxide (Al_2_O_3_) to increase the dielectric, mechanical, and thermal properties of XLPE nanocomposites. The alumina (Al_2_O_3_) nanofiller is well known for its excellent thermal stability and mechanical properties, and its high surface area makes it suitable as a co-nanofiller [13]. Moreover, silane-treated Al_2_O_3_ promotes smooth dispersion state and thus improved compatibility between matrix–filler adhesion [21]. The tensile strength increased 100%; in addition, Young’s modulus of the nanocomposite increased 208%, which contributed to higher stiffness of the nanocomposite resulting in restricting polymer chain mobility. One of the strongest reasons for increasing mechanical properties is the alkyl group of Al_2_O_3_ linked like-a-bridge between Al_2_O_3_ and polymer as -OCH₃ part of trimethoxyoctyl silane chemically bonded to Al_2_O_3_ while the octyl group forms a linkage with the polymer [22].

The hybrid system often exhibits excellent properties, which typically cannot be found in nature because of the unique characteristics of individual nanofillers compared to single nanofillers. Resner et al. [3] studied nanoplatelets–nanotube, whereas Mansor et al. [23] explored spherical–spherical of XLPE nanocomposite that focused on the water treeing phenomenon. Meanwhile, Jose and Thomas [22] researched nanoplatelets–spherical by investigating its mechanical and thermal properties. Most of the properties of hybrid nanofiller tend to increase due to the reinforcing effect and better distribution of hybrid nanofiller into XLPE matrix. Furthermore, the study found that nanotube and spherical nanofillers have a higher tendency to agglomerate at higher concentration (>5 wt.%) than the nanoplatelets, which are well distributed regardless of concentration. Consequently, the hybridising of LDH-Al_2_O_3_ must be studied to investigate the synergistic effect that could enhance the dielectric, mechanical and thermal characteristics of the XLPE nanocomposite as insulation materials.

## 2. Materials and Methods

### 2.1. Materials

The low-density polyethylene (LDPE) with trade name “TITANLENE LDF265YZ” produced by Lotte Chemical Titan (M) Sdn. Bhd., Pasir Gudang, Malaysia was used as the host polymer. The magnesium nitrate hexahydrate (Mg (NO_3_)_2_·6H_2_O) and sodium hydroxide (NaOH) were provided by Fluka and Irganox 1010 from BASF (M) Sdn. Bhd., Pasir Gudang, Malaysia respectively. The aluminium nitrate nonahydrate (Al (NO_3_)_3_·9H_2_O), sodium dodecyl sulfate (SDS), and Al_2_O_3_ nanoparticles with average particle size of <50 nm (TEM) (CAS No.:1344-28-1), trimethoxykt(octyl)silane and dicumyl peroxide (DCP) with average density of 1.56 g/cm^3^ were obtained from Sigma Aldrich, Petaling Jaya, Malaysia, whereas sodium carbonate (NaCO_3_) was supplied by QReC (Asia), Rawang, Malaysia. The deionised water was used as a solvent for preparing all solutions. The chemicals and reagents used in this experiment were of analytical grade and used without further purification.

### 2.2. Preparation of LDH-SDS

The Mg/Al LDH was synthesised with a ratio of 2:1 by the coprecipitation method. The 0.05 mol Mg (NO_3_)_2_·6H2O and 0.025 mol Al (NO_3_)_3_·9H_2_O were dissolved in deionized water (50 mL) and denoted as solution A. Caustic solution was prepared using 0.1 mol NaOH and 0.05 mol Na_2_CO_3_ in 100 mL as solution B, and solution C was 0.05 mol SDS in 100 mL of dissolved deionised water. Solutions A, B, and C were vigorously stirred at room temperature during the preparation. Then, solution A was added drop by drop into solution B and dropwise into solution C, a process that took about 3 h. The solution pH was adjusted and maintained at pH 10.5 ± 0.5. The solution was kept under continuous agitation for 18 h at 65 °C. The solid sample was finally collected using centrifuge at 5000 rpm until it reached pH 7 by washing it with deionised water. The resultant powder was dried in an oven for 12 h at 90 °C and grounded using mortar and pestle to obtain pure Mg/Al powders. The schematic structures of LDH are shown in Figure 1.

### 2.3. Surface Pre-Treatment of Al_2_O_3_

Approximately 11.52 g of Al_2_O_3_ was dispersed into 20:80 (water: 2-propanol) mixture solution at 2750 mL and treated with an ultrasonic bath for 15 min. To promote hydrolysis process, 25% ammonia solution is added in 61.2 mL to the suspension under vigorous stirring together with silane (64.8 mL). The reaction takes 24 h at room temperature and centrifuges at 5000 rpm for 7 min. The Al_2_O_3_ nanoparticles were dried at 80 °C overnight, ground with a pestle and mortar to obtain the fine powder. The silane-Al_2_O_3_ treatment was adopted from Liu et al. [24] with the schematic structures of Al_2_O_3_ shown in Figure 1.

### 2.4. Preparation of Nanocomposite

Figure 2 illustrated the XLPE nanocomposite preparation by a melt mixing method using a Brabender internal mixer at 110 °C with a speed of 60 rpm in 7 min to obtain semi-XLPE. The formulation of XLPE nanocomposite is 0.0, 0.2, 0.5, 0.8, and 1.0 wt.% of hybrid LDH-Al_2_O_3_ nanofiller with a constant ratio of 1:1. All samples contained a composition of constant DCP and Irganox at 1.5 wt.% and 0.25 wt.%, respectively, throughout the preparation. The adding material sequence started with LDPE resin for about 2 min followed by LDH and Al_2_O_3_ at minutes 3 and 4, whereas DCP and Irganox were added later before completing 7 min and these parameters were constant during nanocomposite preparation. The fully crosslinked nanocomposites were obtained using hydraulic press, preheating the samples for 5 min at 120 °C without any pressure to flatten the sample for another 15 min at a temperature of 180 °C with 3.5 tons pressure. The samples were cooled down by cool-pressing for 15 min. All prepared samples were placed in a vacuum drying oven for the degassing process for 36 h at 80 °C prior to testing and characterisation analysis.

### 2.5. Testing and Characterisation

The state of LDH nanofiller was studied using X-ray diffraction studies using a wide angle X-ray diffractometer (WAXD) with Ni-filtered CuK∞ source having a wavelength, λ = 0.154 nm operated at 40 kV and 30 mA with a step size of 0.02 from 2 thetas 1° to 10° (D8 Advance, Bruker AXS, Ettlingen, Germany). Fourier transform infrared spectroscopy (FTIR), Varian 4100 FTIR Excalibur Series instrument, in the attenuated total reflectance, attenuated total reflectance (ATR) mode, and the wavelength of ATR started in the range of 4000–400 cm^−1^, with 32 scans with a resolution of 4 cm^−1^ was used.

For the electrical test, the partial discharge (PD) measurements of the sample were conducted at the high voltage on the disc-like shape sample using a cylindrical-like high voltage electrode with 5 mm thickness of the ideal flat surface. According to the IEC 60270 standard, (a) 1 nF coupling capacitor and measuring impedance were connected parallel to the IEC (b) test containing the nanocomposite samples. The AC breakdown strength tests were conducted according to ASTM D149 standard by submerging the sample into silicon oil between two steel ball-bearing electrodes with a diameter of 6.35 mm. The 50-Hz AC voltage was increased gradually at a rate of 1 kV every 20 s to the sample until the sample experienced breakdown. Three samples were prepared with total points of 15 measurements collected at particular thicknesses for analysis. All the data were analysed using Weibull analysis.

The tensile properties include tensile strength, Young’s modulus, and elongation at break carried out by Zwick with software test expert II at a crosshead speed of 50 mm/min at 25 °C room temperature. The specimen dimensions are Type I according to ASTM D638, with a sample thickness of 3 mm. All measurements were conducted in five replicates, and the value was averaged.

The DSC tests were performed on Mettler Toledo STARe DSC 1 with samples sealed in aluminium pans under a nitrogen atmosphere of 5 mL/min in a temperature range between 25 °C to 300 °C at a heating rate of 10 °C/min. The heat of fusion, ΔHm, was integrated with the DSC endothermic peak while crystallinity, Xc, normalised the heat of fusion to the heat of fusion of 100% crystalline *PE* ΔH*_m_PE* as in Equation (1).
(1)$$\mathrm{Xc}=\;\frac{∆H_{m}}{∆H_{m}PE}$$

The specific enthalpy melting value for 100% crystalline PE was taken as 288 kJ/kg [25].

Morphological feature of the nanocomposite samples was analysed using Carl-Zeiss Supra 35VP FESEM. The fillers were nonconductive samples; thus, the sample needed to be coated in platinum using a platinum sputter coater under vacuum pressure for a time of 1 min at a current of 20 mA and voltage of 1.6 kV to provide electrical conductivity and prevent the surface charge accumulation. The nanocomposite samples were then examined at 10 kV of acceleration voltage.

## 3. Results

### 3.1. Characterisation of LDH Nanofiller

Figure 3a shows the XRD patterns of LDH and LDH-SDS. The diffraction peaks of LDH-SDS are shifted to the lower angle from LDH, indicating the intercalation of SDS. The peaks diffraction pattern is narrow and symmetrical, indicating a high degree of order in the LDH-SDS. The strong diffraction peaks (0 0 3), (0 0 6), and (0 0 9) at 2θ are 3.02°, 12.94°, and 19.49°, respectively, and the LDH-SDS interlayer distance is expanded to 2.923 nm from 0.772 nm of LDH. It is proven that SDS successfully intercalated the interlayer space of LDH. Moreover, a broad reflection in the range of 20°–23° (in the circle) confirmed that the hydrocarbon chains of SDS were intercalated into the interlayer of LDH. In general acknowledgement, the interlayer space is 2.6 nm as SDS anions intercalated into LDH [19].

Figure 3b shows LDH and LDH-SDS composition, and both showed a broad band at 3200 to 3400 cm^−1,^ indicating the stretching mode of hydroxyl group formation in the interlamellar water molecules and the brucite-like layers. The presence of SDS was confirmed by –CH stretching mode 2918, 2852 cm^−1^, while the C–H bending mode band was at 1468 cm^−1^. Meanwhile, the bending mode of the hydroxyl group appeared at 1656 cm^−1^. The typical sulphate absorption bands stretch modes at 1216, 1000, 982, and 824 cm^−1^. The absorption peak for carbonate 1378 cm^−1^ showed greater reduction at LDH-SDS than LDH, indicating SDS replaced carbonate anions into the interlayer [19,26]. The FTIR analysis was performed to certify that SDS was successfully intercalated into the interlayer.

Figure 3c highlights the FESEM morphology for nanofiller is nano-sized with a thickness of less than 30 nm (left) while the width ranges from 60 to 80 nm (right). Meanwhile, the TEM picture shows LDH was stacked one above the other in an orderly and tight manner by strong attractive force within anions interlayer in Figure 3d. This phenomenon was achieved due to a slow and homogeneous precipitation formed of LDH. After SDS ion intercalated inside the LDH nanolayer, the LDH was less orderly stacked on top of each other due to decreasing layers charges. This result has good agreement with XRD analysis.

### 3.2. Dielectric Properties of LDH-Al_2_O_3_ Nanocomposite

#### 3.2.1. Partial Discharge Measurement

Figure 4 shows the comparison of the partial discharge inception voltage (PDIV) and partial discharge extinction voltage (PDEV) on XLPE nanocomposite acquired based on different nanofillers loading. Theoretically, the higher the values of PDIV and PDEV exhibit better-insulating material because it requires a higher voltage level to initiate and extinguish the PD activities. For all samples tested, the trend showed that the values of PDIV and PDEV obtained using cylindrical-shaped high-voltage electrodes exhibited the lowest value at 0.0 wt.%. Introducing LDH-Al_2_O_3_ nanoparticles into the XLPE matrix changes the insulating materials’ PDIV and PDEV. Even though the difference in terms of PDIV and PDEV were not significant from one sample to another, it showed that the amount of fillers had slightly influenced the profile of voltage where the PD signals initiated and extinguished with the percentage of error ranging 0.001 to 0.051.

The highest PDIV and PDEV was at 0.8 wt.% of the filler composition, followed by 0.5, 1.0, and 0.2 wt.%. It was related to the morphological analysis, which illustrated at 0.8 wt.% showed a uniform, smooth, and continuous surface compared to other nanocomposites. This outcome comprehensively reflected the morphological analysis results, which showed that the 0.8 wt.% sample has the largest and strongest interfacial region formed through the formation of interfacial bonds from the nanofillers–polymer surface interactions. At high voltage stresses, the values of PDIV and PDEV are relatively related to the distribution of the electric field on the sample. The effective deep trap is when interfacial region formation is large and strong enough to trap the charges that emitted from the high voltage source. Thus, less agglomeration of nanoparticles would form larger and stronger interfacial regions, which lead to a more effective mechanism in capturing charges [27]. As a result, it may reduce the charge mobility [28], hence reducing the charge transfer rate from the high voltage source to the XLPE nanocomposites under high voltage stress. The results obtained show that the electric field distributed at 0.8 wt.% is better than the other samples due to the minimal local electric field distortion at 0.8 wt.% sample. It seems promising in PD resistance since the values of PDIV and PDEV increase.

Figure 5 shows the phase-resolved partial discharge (PRPD) pattern with the peak charge magnitude for all the PD pulses represented in every single dot. The pattern is asymmetrical on the applied voltage’s positive and negative half cycles. The positive PD pulse count showed it was higher than the negative pulse count for all samples. The PD pulses were captured in the first quadrants, which depicted the phase angle from 0° to 90°, and the third quadrants, which indicated the phase angle from 180° to 270° also represents the discharge that happened on the sample’s surface [29]. The PRPD pattern for all samples tested demonstrated that the type of PD that occurred was surface discharge, which refers to the 50 Hz voltage waveform. The discharge occurred on the surface of samples directly contacted with high voltage electrodes, which is indicated by the characteristics of the charge emitted through the PD activities. The maximum PD charges were extracted and are presented in Figure 6 accordingly.

Figure 6 shows the maximum positive and negative PD magnitude released from the PD activities that occurred on the XLPE nanocomposites at different weight percentages of LDH-Al_2_O_3_ nanoparticles. The presence of LDH-Al_2_O_3_ nanoparticles within the XLPE matrix affected its PD resistance, which was stressed under a high electric field. Apparently, the PD magnitude has changed in each XLPE nanocomposite formulation. The PD magnitude is considered one of the comprehensive parameters in interpreting PD activities. Hence, this parameter is taken into account in determining the PD resistance for different XLPE nanocomposite formulations. The highest PD magnitude released from the PD activities was shown on the unfilled XLPE with 2656 pC positive charge magnitude. Moreover, the highest negative PD charge was exhibited by the PD activities that occurred on the unfilled XLPE with 1245 pC of charge emitted. The non-existent hybrid nanofiller appears to be least resistant against PD attacks, as shown by the highest charge produced compared to the XLPE nanocomposite.

Meanwhile, at 0.2 wt.%, the PD magnitude reduced to 23%. With further addition of 0.5 and 0.8 wt.% LDH-Al_2_O_3_ nanofillers, the PD magnitude has gradually decreased, recorded at 1776, −761 pC and 1384, −638 pC for positive and negative PD magnitude, respectively. Furthermore, 0.2 and 0.5 wt.% were considered insufficient weight percentages of nanofillers due to poor interfacial region, and the charge capturing mechanism tends to be less effective as the samples are stressed under a high electric field. Thus, the PD resistance was not significantly affected because the charges released from the external electrical sources were easy to move around on the sample’s surface and formed ionised channels.

The most effective loading occurred at 0.8 wt.% since it showed the lowest PD magnitude compared to the other weight of nanofillers. This result agreed with the outcomes from the FESEM analysis that showed 0.8 wt.% was the optimum formulation of XLPE nanocomposites, exhibiting well-distributed nanofillers compared to 1.0 wt.% and higher intensity of interfacial bonds formed through the nanofiller–polymer surface interactions compared to 0.2 and 0.5 wt.%. Therefore, the formation of large and robust interfacial regions leads to 0.8 wt.% having higher PD resistance. Unfortunately, 1.0 wt.% led to a significant increase in the PD magnitude recorded at 1523 and −731 pC. These results were due to the occurrence of increased nanofiller loading in the XLPE matrix. Higher filler concentration contributed to space charge trapping that occurred in XLPE, which became bigger [29]. The poor distribution of the nanoparticles is attributable to the incompatible interfaces between nanoparticles and polymer matrix, which eventually lead to the larger PD magnitude.

The addition of nanofillers in the XLPE nanocomposites create a wall by nano-sized filler arrangements in the host polymer, and it performed as a resistance to electron flow between two electrodes during the electrical stress. Therefore, it has been indicated that the XLPE sample with nanofillers had better PD-resistant insulation than the unfilled XLPE. The nanofillers serve as a barrier on the surface sample against PD attack. An indication supported by [30] confirmed that the addition of nanofillers enhanced the ability of XLPE nanocomposites to withstand surface degradation due to PD. Furthermore, the PRPD results show that the occurrences of PD pulses at 0.8 wt.% is the optimum amount of LDH-Al_2_O_3_ nanoparticles to incorporate in the XLPE matrix.

Figure 7 shows results obtained from PD numbers for XLPE nanocomposite. The number of PD pulses was 38751 for the unfilled XLPE, indicating the highest number of PD pulses compared to other XLPE nanocomposite samples. The PD numbers were reduced using 5.7, 8.6, and 22.5% of unfilled XLPE as weight percentages of LDH-Al_2_O_3_ nanoparticles were filled in the XLPE matrix at 0.2, 0.5, and 0.8 wt.%, respectively. Then, the PD pulse count increased to 1.0 wt.% by 8.3% at 32764 and not more than 0.5 wt.% PD pulse count. It is clear that 0.8 wt.% showed the most effective formulation of XLPE nanocomposites with the lowest PD numbers, 30,027. The LDH-Al_2_O_3_ nanoparticles restricted the electron flow between two electrodes when the insulating materials were subjected to electrical stress. Therefore, the samples of XLPE with nanofillers have better PD resistance than the unfilled XLPE. Again, the nanofillers serve as a barrier on the surface sample against PD attack. These findings were supported by Awan et al. [30] and Chandrasekar et al. [31] who confirmed that the addition of nanofillers could enhance the ability of XLPE nanocomposites to withstand surface degradation due to PD. The trend of PD characteristics for PDIV, PDEV, PD magnitudes, and PD numbers have showed the same trend, which indicated that the most effective formulation was sample 0.8 wt.%, followed by sample 1.0 wt.%, and sample 0.5 wt.%. The unfilled sample showed the least resistance against PD attacks.

#### 3.2.2. AC Breakdown Voltage

Figure 8 represents the Weibull probability plot comparing the AC breakdown strength of the XLPE nanocomposites containing 0.2 to 1.0 wt.% of LDH-Al_2_O_3_ nanoparticles. The 0.0 wt.% showed the lowest AC breakdown strength with 169.38 kV/mm. Introducing LDH-Al_2_O_3_ nanoparticles into the XLPE matrix increased the AC breakdown strength of the insulating materials. The breakdown strength increased to 180.50 kV/mm at 0.2 wt.% nanoparticles distributed into the XLPE matrix, which indicates an improvement of 6.6% from the non-existence of hybrid nanofiller; the AC breakdown strength of the nanocomposites of each loading of LDH-Al_2_O_3_ nanofillers increases as long it exists in the nanocomposites. It showed the enhancement of AC breakdown strength up to 11.8% and 15.6% at 0.5 and 0.8 wt.%, respectively. Through these improvements, it was found that the highest AC breakdown strength was 0.8 wt.% XLPE nanocomposites of LDH-Al_2_O_3_ nanoparticles with 195.82 kV/mm, followed by 0.5 and 0.2 wt.% with 189.38 kV/mm and 180.50 kV/mm, respectively. However, the breakdown strength has slightly reduced to 173.97 kV/mm as the loading of hybrid nanofillers increased to 1.0 wt.%. The results align with the previous researchers, Said et al., who also found that the AC breakdown strength of XLPE has improved with the presence of nanoparticles within the XLPE matrix [32].

Based on these findings, the optimum loading was determined to be 0.8 wt.% with the highest AC breakdown strength. It could be due to the compatible nanofiller–polymer surfaces leading to improved filler distribution in XLPE matrices. It related to the distribution of LDH-Al_2_O_3_ nanoparticles within the XLPE matrix, which illustrated that 0.8 wt.% showed a smoother cross-section surface than other XLPE nanocomposites. The local electric field would be distributed more uniformly on the samples by forming interfacial bonds and better distribution of nanofillers [33], and the charges pulled under the influence of the electric field would be trapped on the interfacial regions. Therefore, charge mobility or charge transfer rate would reduce, as a result improving the AC breakdown strength of the XLPE nanocomposites [28].

From the results, the incorporation of LDH-Al_2_O_3_ nanoparticles has increased the AC breakdown strength, and this suggests that the nanoparticles act as electron scavengers in the insulation material under electrical stresses. The electron scavenger mechanism helps in capturing the fast electrons liberated from the external source, reducing the streamer propagation process and consistency [34]. Since the large interfacial regions between nanofillers–polymer contributed to increasing trap charge carriers and the ability to suppress electrons by being trapped on the nanoparticles surface, a reduction in mobility charge carriers occurred, thus improving the AC breakdown strength. The dielectric strength improved due to more time and energy needed to stimulate the charge carriers in forming conduction channels [35], as also highlighted by Montanari et al. [36]. The 1.0 wt.% were considered too high due to the shorter particle–particle distance at higher loading of hybrid nanofillers and tended to cause the nanoparticles to overlap and stick together via the attraction of van der Waals forces [37].

### 3.3. Tensile Properties of LDH-Al_2_O_3_ Nanocomposite

Table 1 shows the mechanical performance of XLPE/LDH-Al_2_O_3_ nanocomposite on tensile properties. The tensile properties are influenced by the content and distribution of incorporation LDH-Al_2_O_3_ nanofillers to XLPE matrix as well as interfacial interaction between XLPE and LDH-Al_2_O_3_ nanoparticles. The tensile strength, Young’s modulus, and elongation at break initially increased and subsequently decreased as further addition of hybrid nanofillers to the XLPE matrix. It is reported in various studies that all samples of the nanocomposite have better tensile properties than unfilled XLPE [38,39,40]. This study ascertained an enhancement of tensile strength with increase in the LDH-Al_2_O_3_ nanofillers content from 0.0 to 0.8 wt.% and slight decrease at 1.0 wt.%. The reinforcing effect and a better dispersion are the main contributions that influenced the quality of the XLPE/LDH-Al_2_O_3_ nanocomposite tensile properties and the molecular chain bonds rarely broken [41].

It is intended that the amount of nanofillers reduce the free volume of the XLPE/LDH-Al_2_O_3_ nanocomposites as well as provide a well-distributed morphology and smoother fracture surfaces. The nano-sized particles provided a large interfacial interaction between LDH and Al_2_O_3_ to XLPE matrix. LDH is suggested to reduce the gaps between Al_2_O_3_ and is fairly connected through the matrix, which creates a network morphology [42]. Hence, LDH and Al_2_O_3_ nanoparticles were forming net structures in the XLPE matrix, improving the structural stability and enhancing the deformation resistance and shear resistance [40]. In the meantime, there was slight reduction in tensile strength at 1.0 wt.% but the value is still higher than 0.0 and 0.2 wt.% of XLPE/LDH-Al_2_O_3_ nanocomposites.

Meanwhile the enhancement in Young’s modulus could be attributed to the synergistic stabilisation of LDH and Al_2_O_3_ that involves the formation of network morphology and the reinforcing effect of the LDH-Al_2_O_3_ on the nanocomposites. Lei et al. reviewed that hybrid nanocomposites, specifically from metal oxide and mineral fillers, have a much higher modulus than polymer matrices [43]. The addition of nanofillers increased the stiffness of the nanocomposite; consequently, Young’s modulus also increased. The stiffness of the LDH-Al_2_O_3_ nanocomposites changed by crystallinity, which was observed from the DSC results. This improvement is attributed to the layer silicate structure of LDH, which mainly contributed to increasing Young’s modulus due to high aspect ratio with large surface area as compared to Al_2_O_3_ nanofillers. As a result, the capability of stress transfer was higher across the reinforcement and nanocomposites; thus, the applied stress is disseminated by the interface more evenly [44]. Moreover, the tendency to interconnect and form a network structure of LDH is larger than in Al_2_O_3_ nanofillers [39].

The increasing of elongation at break values of XLPE/LDH-Al_2_O_3_ nanocomposites due to incorporation of a low number of nanoparticles into the nanocomposite improved the interaction between the molecules by slipping without breaking the samples due to the nanoparticles being reinforced and oriented along the stress direction resulting in higher elongation at breaks. However, further addition of LDH-Al_2_O_3_ nanofillers to the XLPE matrix subsequently increased the restriction of mobility on polymer chains; thus, the elongation at break showed a decrement curve. Notably, this situation is expected in most polymer nanocomposites [32].

The small amount of LDH-Al_2_O_3_ nanofiller provides a huge total interface area and increases reinforcement efficiency. The reinforcing effect of the small amount of nanoparticle loadings has huge specific surface area and dramatically larger total interface area for reinforcement work efficiency. It is shown that, at lower weight percentage, the addition of LDH-Al_2_O_3_ nanofiller in XLPE matrix increases surface interaction bonding between the molecules. At the high amount of LDH-Al_2_O_3_ nanofiller, tensile properties showed a gradual drop. These experimental results are attributed to the reinforcing effect of the nanoparticles in which a higher number of nanoparticles reduces the reinforcing effect by the poor dispersion, agglomeration of nanoparticles, and large volumes of voids. These agglomerated nanoparticles occur due to the filler–filler interaction being higher than filler–matrix interaction and acting as stress concentration in nanocomposites. The slight volume of voids at the interface or trapped in a cluster would make the molecule move freely; thus, the polymer chains’ mobility slipped past one another and later decreased the tensile performance [45].

Generally, the addition of nanofillers to most polymer nanocomposites is expected to reduce the values because of the restriction in chain mobility caused by poor interconnection between nanofillers and polymer chains [45]. However, the results obtained at 0.8 wt.% significantly increased the three key mechanical properties, for instance tensile strength, Young’s modulus and elongation at break. The improvement in mechanical properties have been correlated with the surface fracture of nanocomposite in Figure 9. Moreover, as the nanofillers–matrix are located close to each other, the nanofillers surface needs higher temperature to induce the motion whereas the matrix which is unaffected with nanofillers surface properties is unchanged [46].

### 3.4. Morphology of LDH-Al_2_O_3_ Nanocomposite

Figure 9 shows cross-sectional morphologies of the XLPE/LDH-Al_2_O_3_ nanocomposite using FESEM. The nanocomposite displays good distribution and shows no obvious visible agglomeration of hybrid LDH-Al_2_O_3_ nanofiller at different weight percentages in XLPE to a large extent. From the cross-section, surface fracture of the nanocomposite showed the tendency of nanofillers to aggregate is low. This situation is explained by interfiller interaction being low compared to nanofiller–matrix interaction due to the different surface characteristics and different surface energies that belong to hybrid nanofillers. The FESEM images showed that hybrid nanofillers have great potential to resolve the agglomeration of nanofillers and the nanocomposite surface was observed to become rougher with the addition of hybrid nanofiller into XLPE nanocomposites.

### 3.5. Differential Scanning Calorimetry of XLPE/LDH-Al_2_O_3_ Nanocomposite

Table 2 provides the thermal performance of XLPE/LDH-Al_2_O_3_ nanocomposite based on crystallisation temperature (Tc), melting temperature ™, enthalpy (ΔHm), and crystallinity (Xc) of nanocomposite. Both Tm and Tc increased slightly as a result of nanofiller content of all nanocomposite samples. The Tm and Tc move toward the high temperature direction, from 107 °C to 109 °C and 90 °C to 94 °C, respectively, with the introduction of LDH-Al_2_O_3_ nanofillers. The temperature increment is due to increased interaction between the XLPE matrix and LDH-Al_2_O_3_ nanofillers, which lead from restricted mobility of XLPE chains to relaxation of the nanocomposite system at higher temperature. Another reason for increment of about 2 °C is attributed to the introduction of LDH-Al_2_O_3_ is that it provides better networking to improve thermal properties [47]. Donghe and Qingyue concluded the introduction of nanoparticles in XLPE showed two main parameters: first, an increase in the melting point; second, the ability of crystallisation to slow down due to restriction in movement of molecular chain segment [40,48].

Based on DSC thermograms in Figure 10a,b, the endothermic recorded during the second heating cycle and the exothermic curve was recorded cooling from the melt of the first heating cycle, respectively. The degree of crystallinity, which can be calculated using the melting enthalpy with the specific enthalpy melting value for 100% crystalline PE, was taken as 288 kJ/kg. Note that the slight increase in Tc and Tm from unfilled XLPE to XLPE nanocomposite ensures that the crystalline region melts and enhances mechanical properties at relative temperatures [49]. This finding is in agreement with Thomas et al. [50] stating that DSC results of XLPE nanocomposites had comparable results on Tc and Tm indicating a low effect of nanofillers on XLPE phase transformations. However, after analysing the value of the Xc, it can be seen that the incorporation of LDH-Al_2_O_3_ to XLPE caused the Xc to increase. It was also that with the addition of LDH-Al_2_O_3_ content, an explicit decrease in the Xc was observed but still higher than the unfilled XLPE [3].

## 4. Conclusions

The incorporation of hybrid LDH-Al_2_O_3_ nanofiller into XLPE matrix has successfully improved the dielectric properties of PD and AC breakdown voltage without sacrificing the mechanical properties of the nanocomposites. Based on the amount of hybrid nanofiller, the 0.8 wt.% showed the potential hybrid loading of XLPE/LDH-Al_2_O_3_ nanocomposites. The dielectric properties were enhanced, with PD magnitude reducing to +1384, −638 and the PD number being reduced to 30,027, whereas the mechanical properties exhibit the highest tensile strength, Young’s modulus, and elongation at break of 18.1 MPa, 205 MPa, and 633%, respectively. Moreover, the formation of a strong interphase region was promoted by the balance effect of morphological features of the hybrid network of each component with no visible agglomeration. It was found that increasing the crystallinity slightly improved the thermal properties of the hybrid nanocomposite in this study. The outcomes of this work provide further guidance for the design of high-voltage direct current insulation materials.

## Figures and Tables

**Figure 1 polymers-15-01702-f001:**
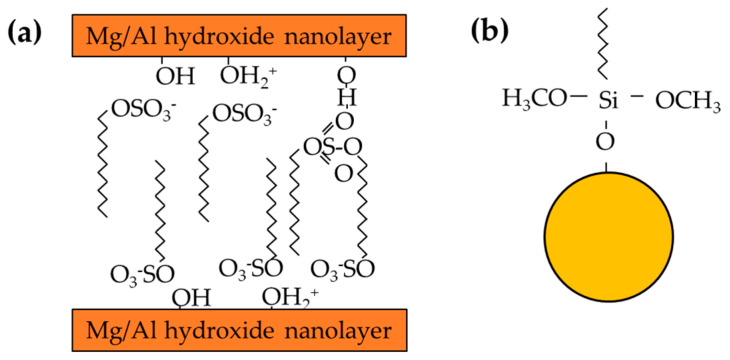
The structures of (**a**) LDH−SDS, (**b**) silane Al_2_O_3_.

**Figure 2 polymers-15-01702-f002:**
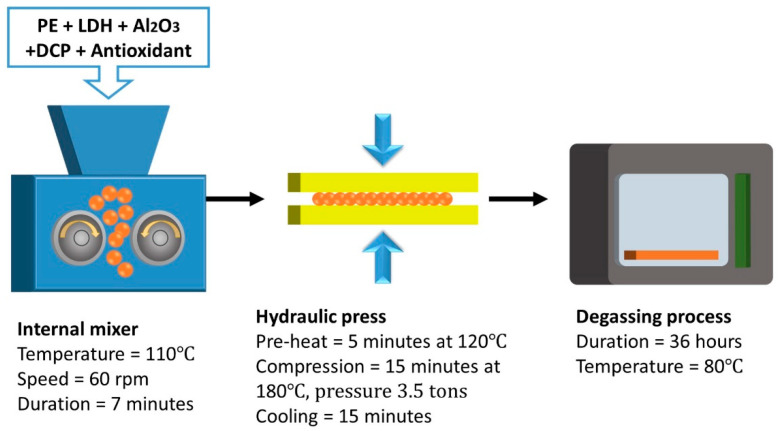
Schematic of the preparation of XLPE/LDH−Al_2_O_3_ nanocomposites.

**Figure 3 polymers-15-01702-f003:**
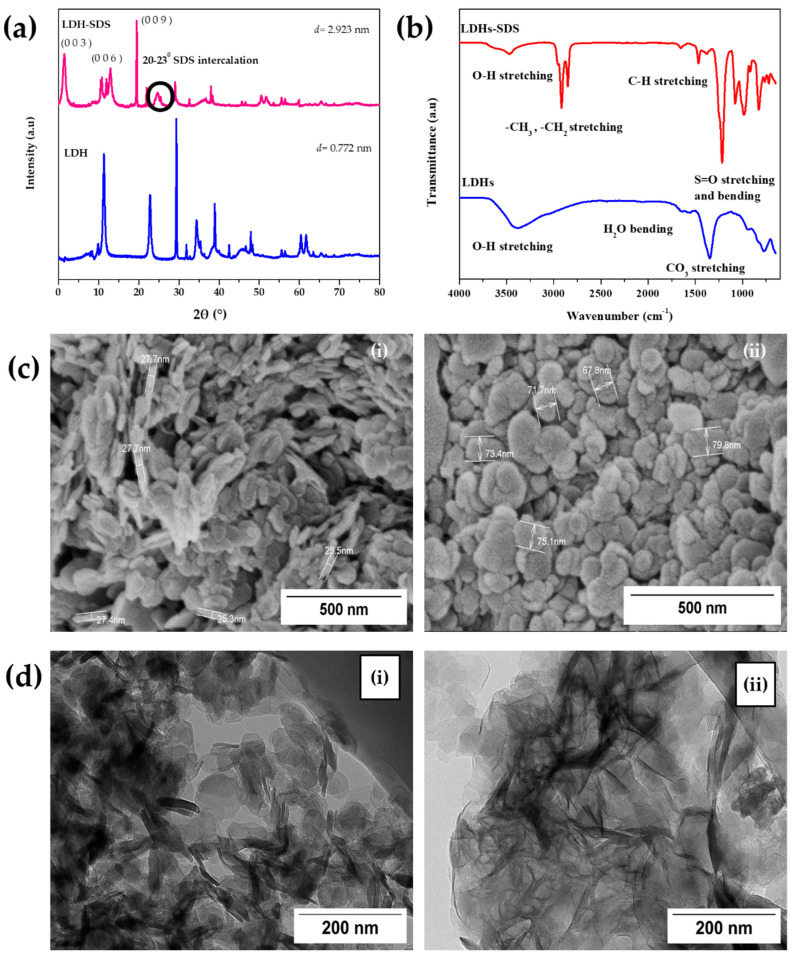
The nanofiller characterization on (**a**) XRD pattern of LDH and LDH−SDS, (**b**) FTIR spectra of LDH and LDH−SDS, (**c**) FESEM image (**i**) LDH thickness and (**ii**) width, (**d**) TEM image of (**i**) LDH and (**ii**) LDH−SDS nanofiller.

**Figure 4 polymers-15-01702-f004:**
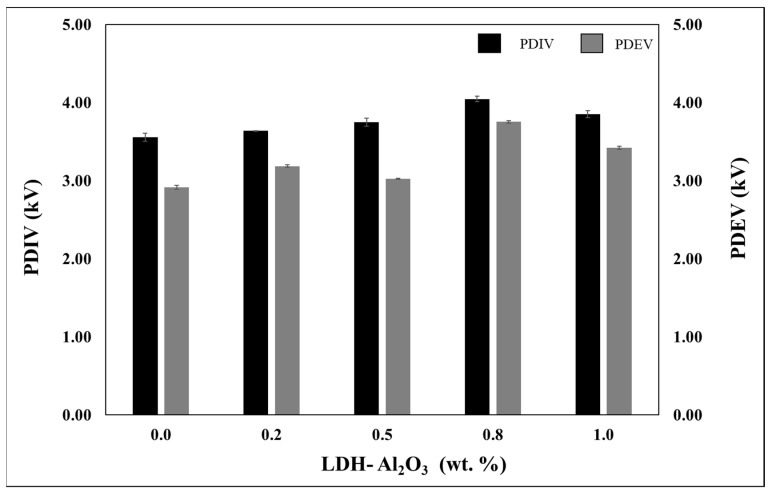
PDIV and PDEV of XLPE/LDH−Al_2_O_3_ nanocomposite.

**Figure 5 polymers-15-01702-f005:**
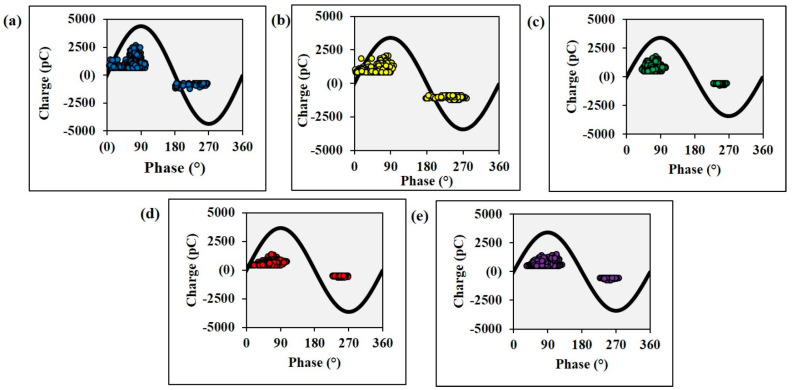
PRPD patterns of the XLPE/LDH−Al_2_O_3_ nanocomposites at (**a**) 0.0 wt% (**b**) 0.2 wt.% (**c**) 0.5 wt.% (**d**) 0.8 wt.% and (**e**) 1.0 wt.%.

**Figure 6 polymers-15-01702-f006:**
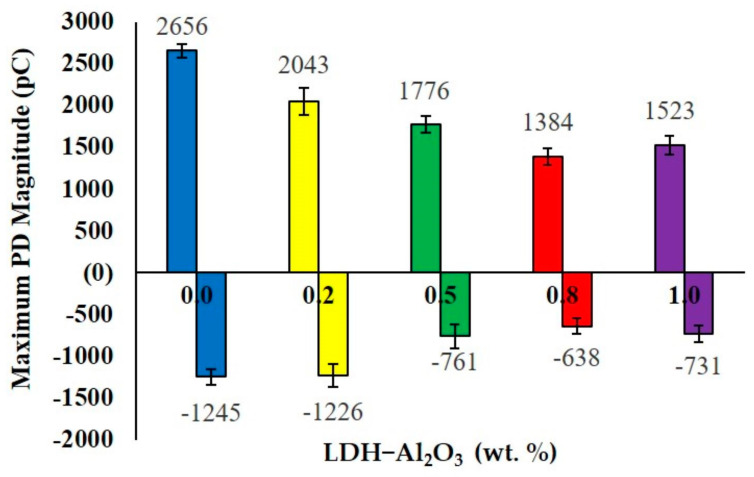
The comparison of maximum PD magnitude of XLPE/LDH−Al_2_O_3_ nanocomposites at varying the weight percentage of nanofillers.

**Figure 7 polymers-15-01702-f007:**
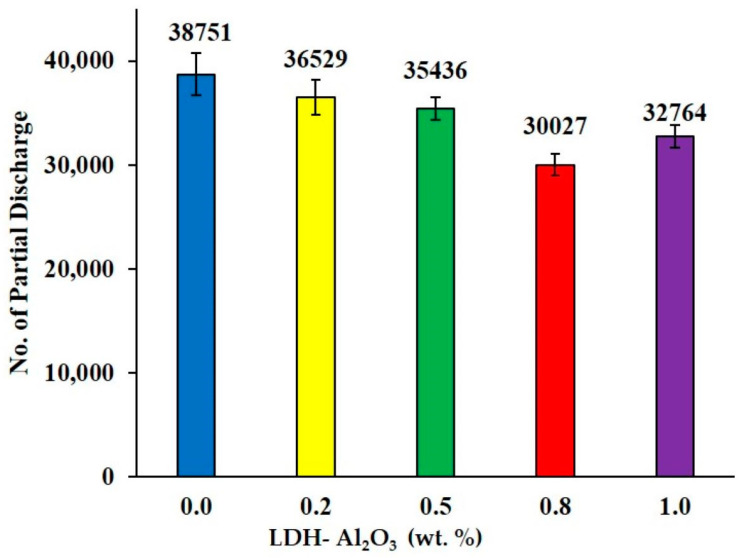
The total PD numbers of XL PE/LDH−Al_2_O_3_ nanocomposites at varying the weight percentage of nanofillers.

**Figure 8 polymers-15-01702-f008:**
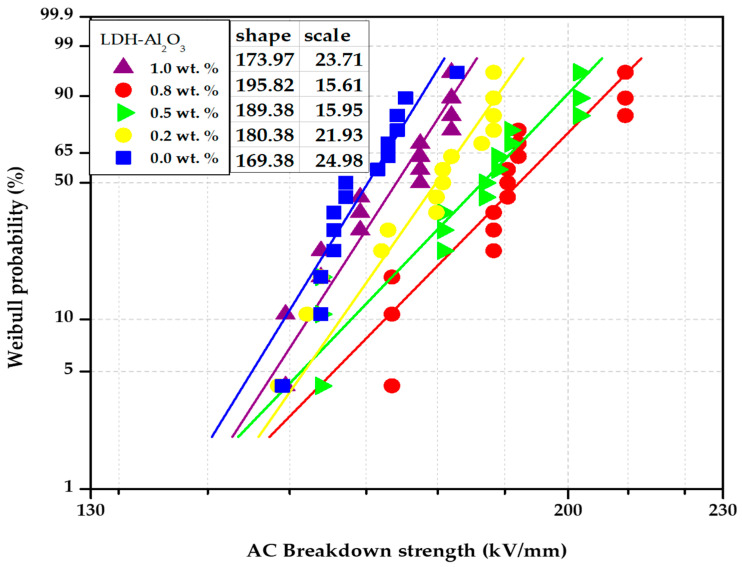
The Weibull analysis plot comparing the AC breakdown strength of XLPE/LDH-Al_2_O_3_ nanocomposites.

**Figure 9 polymers-15-01702-f009:**
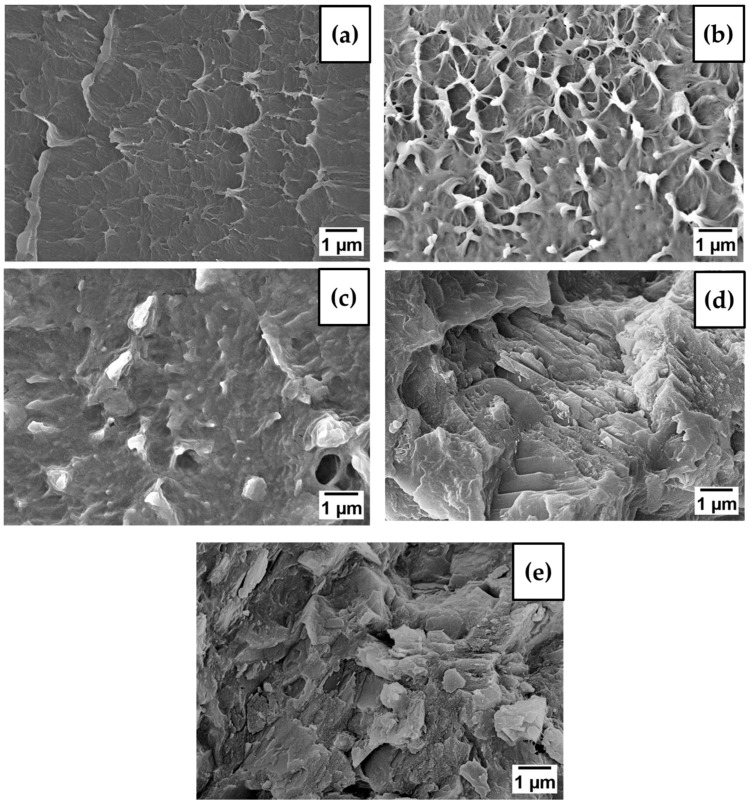
The FESEM image of XLPE/LDH-Al_2_O_3_ nanocomposite at (**a**) 0.0 wt% (**b**) 0.2 wt.% (**c**) 0.5 wt.% (**d**) 0.8 wt.% and (**e**) 1.0 wt.%.

**Figure 10 polymers-15-01702-f010:**
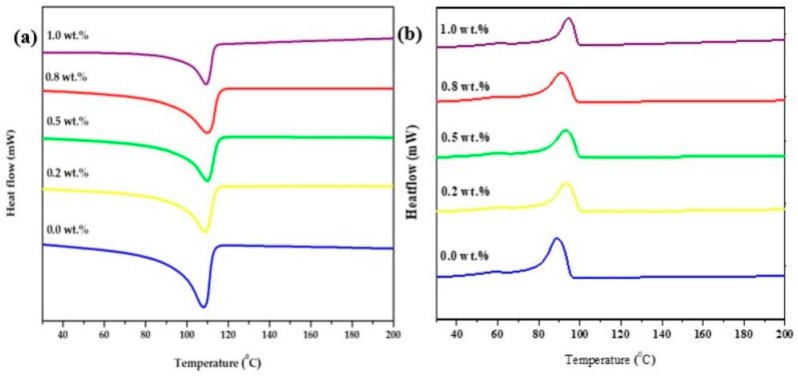
The DSC thermograms of XLPE/LDH-Al_2_O_3_ nanocomposites (**a**) 2nd heating cycle (**b**) cooling cycle.

**Table 1 polymers-15-01702-t001:** This mechanical properties of XLPE/LDH-Al_2_O_3_ nanocomposite.

LDH-Al_2_O_3_ (Wt.%)	Mechanical Performance of the Nanocomposite		
	Tensile Strength (MPa)	Young’s Modulus (MPa)	Elongation at Break (%)
0.0	15.5 ± 0.7	140.8 ± 21.0	484 ± 16
0.2	16.9 ± 0.9	153.7 ± 12.6	510 ± 29
0.5	17.8 ± 0.4	174.3 ± 23.5	581 ± 15
0.8	18.1 ± 0.6	205.0 ± 16.6	633 ± 19
1.0	17.2 ± 0.8	180.4 ± 20.5	563 ± 18

**Table 2 polymers-15-01702-t002:** The DSC analysis of XLPE/LDH-Al_2_O_3_ nanocomposite.

LDH-Al_2_O_3_ (Wt.%)	Thermal Performance of Nanocomposite			
	Crystallisation Temperature (°C)	Melting Temperature (°C)	Enthalpy (J/g)	Crystallinity (%)
0.0	90.5	107.8	99.31	34.5
0.2	93.7	109.7	124.3	43.2
0.5	93.5	109.5	125.9	43.7
0.8	93.2	109.3	132.8	46.1
1.0	94.5	109.1	101.7	35.3

## Data Availability

The data presented in this study are available upon request from the corresponding author.

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
