# Peer review of "Dielectric, Mechanical, and Thermal Properties of Crosslinked Polyethylene Nanocomposite with Hybrid Nanofillers"

_polymers, 2023, doi:10.3390/polym15071702_

Round 1

Reviewer 1 Report

The paper presents the study about dielectric, mechanical, and thermal properties of XLPE nanocomposite with hybrid nanofillers. Authors studied incorporation amount of hybridising LDH with Al2O3 nanofiller into the XLPE matrix. They investigated the influence of hybrid nanofiller content on the dielectric, mechanical and thermal properties of the nanocomposite. Some advantages of proposed results was to improve the dielectric, mechanical and thermal properties, including partial discharge resistance, AC breakdown strength, tensile properties.

Dear author, thank you very much for interesting paper about the impact of nanocomposite on many important properties of XLPE, used as high voltage insulation in case of power cables. I put some comments and questions.

Comments:

1. The introduction chapter is well organized. Used references are correct. Authors describes main role of XLPE as insulation material used in high voltage power cables. They also explain previous impact of nano particles on XLPE properties.

2. Materials chapter also explains all necessary information. anyway, I could not find information about size used nano particles of Al2O3. Please explain why or complete.

3. Fig.4 – inception voltage of PD increases with the increase of nano, but the increase is very small. If we consider some measurements error, probably the increase will be very small. What is advantage in this moment?

4. Fig.5 – probably the same comments are to fig.5 as for fig.4.

5. In case of Fig.6 there is big improvement of proposed solutions, I mean the use of nano particles to improve the value of PD charges. The same comment is for fig.7.

6. In case of fig.8. – AC breakdown strength, there is some progress, but also there is some extremum of the nano particles use. Please explain the maximum from physical point of view. The same comments are in case of mechanical properties, see table 1, and see table 2. Please explain the maxima of the nano particles use.

Author Response

We would like to thank you for the opportunity to revise and resubmit this manuscript (Polymers-2233650). We found the reviewer’s comments to be insightful in improving the article. To address the comments, we have previously listed all of the response for the betterment of the manuscript. We have managed to reply all of the comments and suggestion by the reviewer. We have included replied to the comments in which we address the changes in red font in the revised manuscript. More than that, the manuscript has been extensively revised by a native English-speaking proofreader.

Thank you.

Reviewer 2 Report

The authors have presented a polyethylene nanocomposite with hybrid nanofillers to improve the electrical properties of crosslinked polyethylene without degrading the mechanical properties. The hybrid fillers are shown to improve the properties in general due to good interfacial interactions with the polyethylene matrix. Overall, this is a good study but needs significant text edits to improve the readability for publication in Polymers.  

Major Comments (Major scientific and technical concerns)

1. The enhancements (numbers or %) should be included in the abstract.

2. In section 3.2.2 line 348, it is stated that the enhancement due to 0.5 and 0.8 wt% loading is 5.1 and 3.22% respectively however this is lower than 6.6% enhancement due to 0.2 wt% loading. The enhancement % for 0.5 and 0.8 wt% should be re-calculated and corrected.

3. In table 1, the mechanical properties presented do not have a standard deviation. Without at least three samples these numbers do not carry statistical significance. Please test more samples and update the table.

4. The conclusion should also include numerical values of the enhanced properties.

Minor Comments (Minor concerns)

1. Irregular spacing should be corrected throughout.

Author Response

(The authors gave the same response as above.)
